# Improved protein structure refinement guided by deep learning based accuracy estimation

Naozumi Hiranuma[1,2,4], Hahnbeom Park[1,4], Minkyung Baek [1], Ivan Anishchenko [1], Justas Dauparas[1] & David Baker [1,3 ✉]

We develop a deep learning framework (DeepAccNet) that estimates per-residue accuracy and residue-residue distance signed error in protein models and uses these predictions to guide Rosetta protein structure refinement. The network uses 3D convolutions to evaluate local atomic environments followed by 2D convolutions to provide their global contexts and outperforms other methods that similarly predict the accuracy of protein structure models. Overall accuracy predictions for X-ray and cryoEM structures in the PDB correlate with their resolution, and the network should be broadly useful for assessing the accuracy of both predicted structure models and experimentally determined structures and identifying specific regions likely to be in error. Incorporation of the accuracy predictions at multiple stages in the Rosetta refinement protocol considerably increased the accuracy of the resulting protein structure models, illustrating how deep learning can improve search for global energy minima of biomolecules.

[1] Department of Biochemistry and Institute for Protein Design, University of Washington, Washington, WA, USA. [2] Paul G. Allen School of Computer Science & Engineering, University of Washington, Washington, WA, USA. [3] Howard Hughes Medical Institute, University of Washington, Washington, WA, USA. [4] These authors contributed equally: Naozumi Hiranuma, Hahnbeom Park. ✉email: dabaker@uw.edu

D istance prediction through deep learning on amino acid co-evolution data has considerably advanced protein structure prediction[1–3]. However, in most cases, the predicted structures still deviate considerably from the actual structure[4]. The protein structure refinement challenge is to increase the accuracy of such starting models. To date, the most successful approaches have been with physically based methods that involve a large-scale search for low energy structures, for example with Rosetta[5] and/or molecular dynamics[6]. This is because any available homology and coevolutionary information is typically already used in the generation of the starting models.

The major challenge in refinement is sampling; the space of possible structures that must be searched through even in the vicinity of a starting model is extremely large[5,7]. If it were possible to accurately identify what parts of an input protein model were most likely to be in error, and how these regions should be altered, it should be possible to considerably improve the search through structure space and hence the overall refinement process. Many methods for estimation of model accuracy (EMA) have been described, including approaches based on deep learning such as ProQ3D (based on per-residue Rosetta energy terms and multiple sequence alignments with multilayer perceptrons[8]), and Ornate (based on 3D voxel atomic representations with 3D convolutional networks[9]). Non-deep learning methods such as VoroMQA compare a Voronoi tessellation representation of atomic interactions against precollected statistics[10]. These methods focus on predicting per-residue accuracy. Few studies have sought to guide refinement using deep learning based accuracy predictions[11]; the most successful refinement protocols in the recent blind 13th Critical Assessment of Structure Prediction (CASP13) test either utilized very simple ensemble-based error estimations[5] or none at all[12]. This is likely because of the low specificity of most current accuracy prediction methods, which only predict which residues are likely to be inaccurately modeled, but not how they should be moved, and hence are less useful for guiding search.

In this work, we develop a deep learning based framework (DeepAccNet) that estimates the signed error in every residue–residue distance along with the local residue contact error, and we use this estimation to guide Rosetta based protein structure refinement. Our approach is schematically outlined in Fig. 1.

## Results

**Development of improved model accuracy predictors**. We first sought to develop model accuracy predictors that provide both global and local information for guiding structure refinement. We developed network architectures that make the following three types of predictions given a protein structure model: per-residue $C_\beta$ local distance difference test ($C_\beta$ l-DDT) scores which report on local structure accuracy[13], a native $C_\beta$ contact map thresholded at 15 Å (referred to as "mask"), and per-residue-pair distributions of signed $C_\beta$–$C_\beta$ distance error from the corresponding native structures (referred to as "estograms"; histogram of errors); $C_\alpha$ is taken for GLY. Rather than predicting single error values for each pair of positions, we instead predict histograms of errors (analogous to the distance histograms employed in the structure prediction networks of refs. [1–3]), which provide more detailed information about the distributions of possible structures and better represent the uncertainties inherent to error prediction. Networks were trained on alternative structures ("decoys") with model quality ranging from 50 to 90% in GDT-TS (global distance test—tertiary structure)[14] generated by homology modeling[15], trRosetta[1], and native structure perturbation (see Methods). Approximately 150 decoy structures were generated for each of 7307 X-ray crystal structures with

resolution better than 2.5 Å lacking extensive crystal contacts and having sequence identity less than 40% to any of 73 refinement benchmark set proteins (see below). Of the approximately one million decoys, those for 280 and 278 of the 7307 proteins were held-out for validation and testing, respectively. More details of the training/test set and decoy structure generation can be found in Methods.

The predictions are based on 1D, 2D, and 3D features that reflect accuracy at different levels. Defects in high-resolution atomic packing are captured by 3D-convolution operations performed on 3D atomic grids around each residue defined in a rotationally invariant local frame, similar to the Ornate method[9]. 2D features are defined for all residue pairs, and they include Rosetta inter-residue interaction terms, which further report on the details of the interatomic interactions, while residue–residue distance and angular orientation features provide lower resolution structural information. Multiple sequence alignment (MSA) information in the form of inter-residue distance prediction by the trRosetta[1] network and sequence embeddings from the ProtBert-BFD100 model[16] (or Bert, in short) are also optionally provided as 2D features. At the 1D per-residue level, the features are the amino acid sequence, backbone torsion angles, and the Rosetta intraresidue energy terms (see Methods for details).

We implemented a deep neural network, DeepAccNet, that incorporates these 1D, 2D, and 3D features (Fig. 1a). The networks first perform a series of 3D-convolution operations on local atomic grids in coordinate frames centered on each residue. These convolutions generate features describing the local 3D environments of each of the N residues in the protein. These, together with additional residue level 1D input features (e.g., local torsional angles and individual residue energies), are combined with the 2D residue–residue input features by tiling (so that associated with each pair of residues there are both the input 2D features for that pair and the 1D features for both individual residues), and the resulting combined 2D feature description is input to a series of 2D convolutional layers using the ResNet architecture[17]. A notable advantage of our approach of tying together local 3D residue based atomic coordinate frames through a 2D distance map is the ability to integrate full atomic coordinate information in a rotationally invariant way; in contrast, a Cartesian representation of the full atomic coordinates would change upon rotation, substantially complicating network for both training and its use. Details of the network architecture, feature generation, and training processes are found in Methods.

Figure 2 shows examples of the predictions of DeepAccNet without MSA or Bert embeddings (referred to as "DeepAccNet-Standard") on two randomly selected decoy structures for each of three target proteins (3lhnA, 4gmqA, and 3hixA) not included in training. In each case, the network generates different signed residue–residue distance error maps for the two decoys that qualitatively resemble the actual patterns of the structural errors (rows of Fig. 2). The network also accurately predicts the variations in per-residue model accuracy ($C_\beta$ l-DDT scores) for the different decoys. The left sample from 4gmqA (second row) is closer to the native structure than the other samples are, and the network correctly predicts the location of the smaller distance errors and $C_\beta$ l-DDT scores closer to 1. Overall, while the detailed predictions are not pixel-perfect, they provide considerable information on what parts of the structure need to move and in what ways to guide refinement. Predictions from the variants with the MSA (referred to as "DeepAccNet-MSA") and Bert features (referred to as "DeepAccNet-Bert") are visualized in Supplementary Fig. 1.

We compared the performance of the DeepAccNet networks to that of a baseline network trained only on residue–residue $C_\beta$ distances. The performances of the DeepAccNet networks

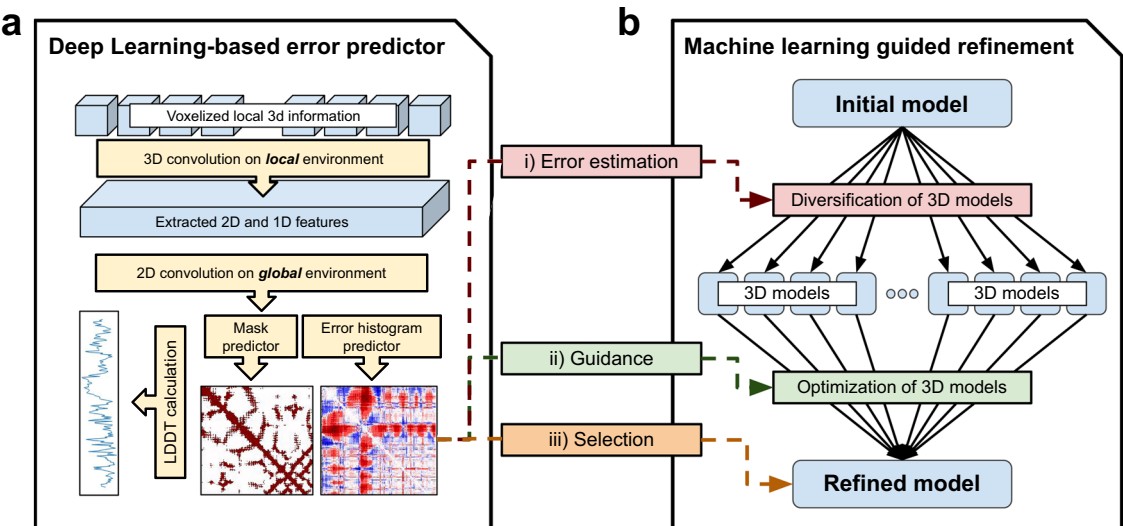

**Fig. 1 Approach overview. a** The deep learning network (DeepAccNet) consists of a series of 3D and 2D convolution operations. The networks are trained to predict (i) the signed $C_\beta$–$C_\beta$ distance error distribution for each residue pair (error histogram or *estogram* in short), (ii) the native $C_\beta$ contact map with a threshold of 15 Å (referred to as mask), (iii) the $C_\beta$ l-DDT score per residue; $C_\alpha$ is taken for GLY. Input features to the network include: distance maps, amino acid identities and properties, local atomic environments scanned with 3D convolutions, backbone angles, residue angular orientations, Rosetta energy terms, and secondary structure information. Multiple sequence alignment (MSA) information in the form of inter-residue distance prediction by the trRosetta network and sequence embeddings from the ProtBert-BFD100 model (or Bert, in short) are also optionally provided as 2D features. Details of the network architecture and features are provided in Methods. **b** The machine learning guided refinement protocol uses the trained neural networks in three ways; the estimated $C_\beta$ l-DDT scores are used to identify regions for more intensive sampling and model recombination, the estimated pairwise error distributions are used to guide diversification and optimization of structure(s), and finally the estimated global $C_\beta$ l-DDT score, which is mean of per-residue values, to select models during and at the end of the iterative refinement process.

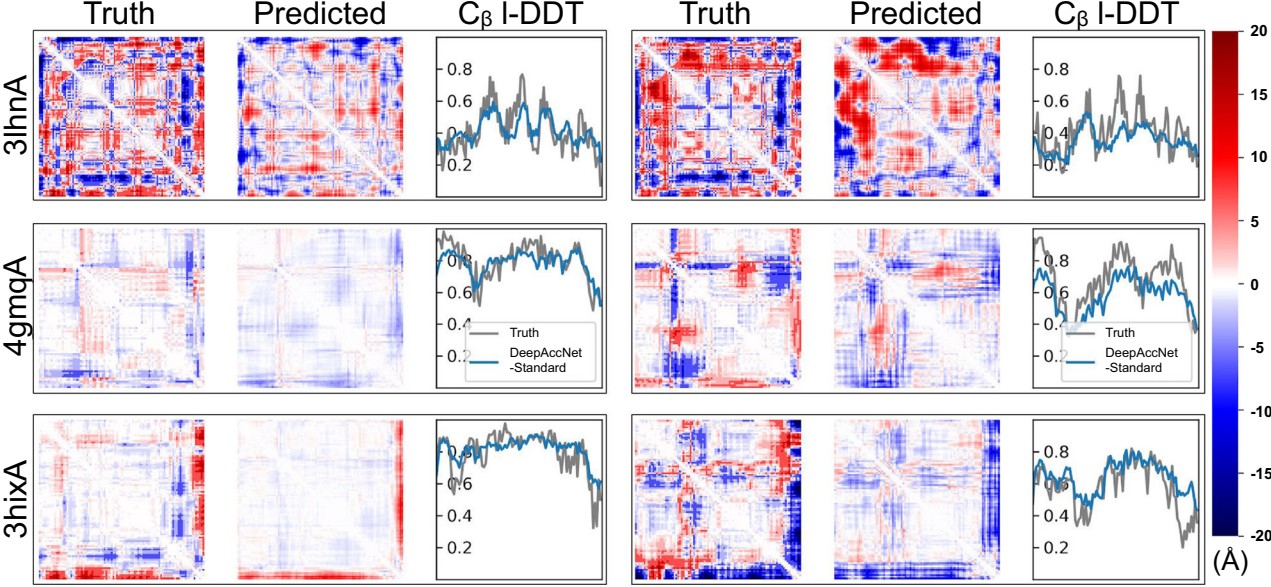

**Fig. 2 Example estograms and $C_\beta$ l-DDT score prediction.** DeepAccNet-Standard predictions for two randomly selected decoys for three test proteins were randomly selected (3lhnA, 4gmqA, 3hixA; size 108, 92, and 94, respectively; black rectangular boxes delineate results for single decoy). The first and fourth columns show true maps of errors, the second and fifth columns show predicted maps of errors, and the third and sixth columns show predicted and true $C_\beta$ l-DDT scores. The i, j element of the error map is the expectation of actual or predicted estograms between residues i and j in the model and native structure. Red and blue indicate that the pair of residues are too far apart and too close, respectively. The color density shows the magnitude of expected errors.

are considerably better on average for almost all the test set proteins (Supplementary Fig. 2a; Fig. 3); they outperform the baseline $C_\beta$ distance model in predicting estograms for residue pairs across different sequence separations and input distances (Supplementary Fig. 2b). The addition of the MSA or Bert information improves overall accuracy particularly for quite

inaccurate models and residues (Supplementary Fig. 2c, d). For all networks, $C_\beta$ l-DDT score prediction performance does not decline substantially with increasing size (Spearman correlation coefficient, or Spearman-r, of $-0.04$ with $p$-value $> 0.05$ for protein size vs. DeepAccNet-Standard performance), but estogram prediction performance clearly declines for larger

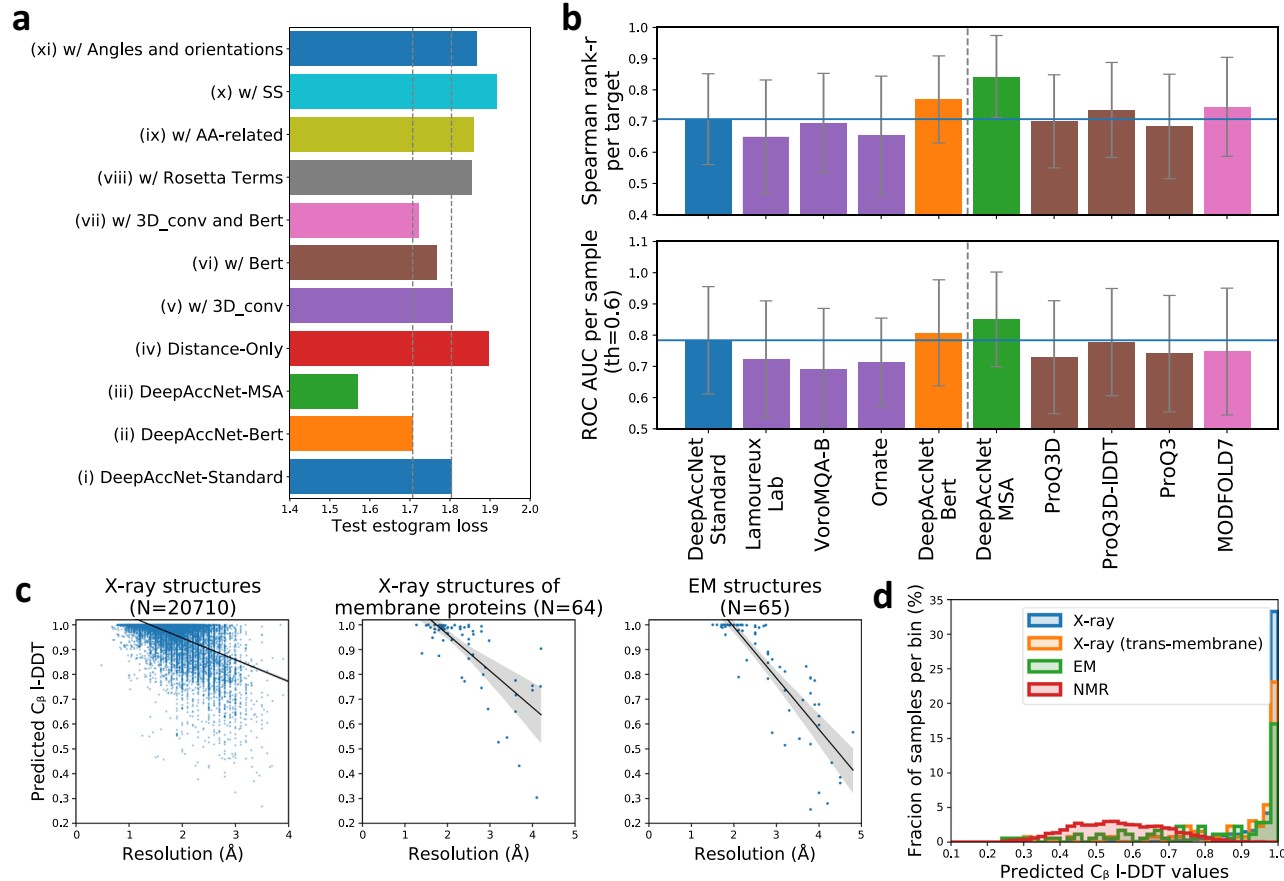

**Fig. 3 DeepAccNet performance. a** Contribution of individual features to network performance; all models include the distance matrix features. Overall, the largest contribution is from the features generated by 3D convolutions on local environments, Bert embeddings, and MSA information. Estogram (cross-entropy) loss values averaged overall decoys for each test protein are shown as one data point. The gray dotted line shows the values from predictors (i) and (ii). **b** Comparison of the performance of single-model accuracy estimation (EMA) methods on CASP13 data. (top) Performance of global accuracy estimation measured by the mean of the Spearman correlation coefficient (r-value) of predicted and actual global l-DDT scores per target protein. (bottom) Performance of local accuracy estimation measured by the mean of area under receiver operator characteristic (ROC) curve for predicting mismodeled residues per sample (all-atom l-DDT < 0.6). The blue horizontal lines show the value of DeepAccNet-Standard. The methods to the left of the dotted line do not use coevolutionary information. Quasi-single EMA method is shown in pink. Error bars show standard deviation. **c** Predicted $C_\beta$ l-DDT by DeepAccNet-Standard correlates with resolution for X-ray structures (left; Spearman-r 0.48 with p-value < 0.0001), X-ray structures of transmembrane proteins (middle; Spearman-r 0.64 with p-value < 0.0001), and cryoEM structures (right; Spearman-r 0.87 with p-value < 0.0001). **d** X-ray structures have higher predicted $C_\beta$ l-DDT values by DeepAccNet-Standard than NMR structures.

proteins (Spearman-r of 0.57 with p-value < 0.00001) (Supplementary Fig. 2e)—for larger proteins with more interactions over long distances, estimating the direction and magnitude of errors is a much harder task while since $C_\beta$ l-DDT scores only consider local changes at short distances, they degrade less with increasing size.

In addition to distance map features, DeepAccNet networks take as input (a) amino acid identities and properties, (b) local atomic 3D environments for each residue, (c) backbone torsion angles and residue–residue orientations, (e) Rosetta energy terms, (f) secondary structure information, (g) MSA, and (h) Bert information. To investigate the contributions of each of these features to network performance, we combined each with distance maps one at a time during training and evaluated performance through estogram cross-entropy loss and $C_\beta$ l-DDT score mean squared error on test sets (Fig. 3a, Supplementary Table 1). Apart from the MSA features, the largest contributions were from the 3D-convolution-based features and the Bert embeddings (compare (v), (vi), and (vii)). There is a statistically significant difference between the network (ii) and (vii), suggesting that the features other than 3D convolution and Bert

help them glue together (p-value < 0.0001 with Wilcoxon signed-rank test for estogram loss between network (ii) and (vii)).

An effective accuracy prediction method should be useful for evaluating and identifying potential errors in experimentally determined structures as well as computational models. We investigated the performance of the network on experimental structures determined by X-ray crystallography, nuclear magnetic resonance spectroscopy (NMR), and electron microscopy (EM) that were not included in the training set (details of the dataset can be found in Methods). The predicted $C_\beta$ l-DDT values by the DeepAccNet variants are close to 1.0 for high-resolution crystal structures, as expected for nearly error free protein structures, and decreases for lower resolution structures (Fig. 3c, left panel for DeepAccNet-Standard, Supplementary Fig. 3 for other variants). A similar correlation between predicted accuracy and resolution holds for X-ray structures of membrane proteins (Fig. 3c, middle panel; Spearman-r 0.64 with p-value < 0.0001) and cryoEM structures (Fig. 3c, right panel; Spearman-r 0.87 with p-value < 0.0001). A list of X-ray structures with low predicted $C_\beta$ l-DDT despite their high experimental resolution is provided in Supplementary Table 2. Many of these are heme-proteins; as the

network does not consider bound ligands, the regions surrounding them are flagged as atypical for folded proteins, suggesting that the network may also be useful for predicting cofactor binding and other functional sites from apo-structures. NMR structures have lower predicted accuracies than high-resolution crystal structures (Fig. 3d, right; Supplementary Fig. 3c, d), which is not surprising given i) they were not included in the training set and ii) they represent solution averages rather than crystalline states.

We compared the DeepAccNet variants to other accuracy estimators (Fig. 3b). As is clear from recent CASP experiments, co-evolution information derived from multiple sequence alignments provides detailed structure information; we include this as an optional input to our network (DeepAccNet-MSA) for two reasons: first, all available homology and coevolutionary information is typically already used in generating the input models for protein structure refinement and second, in applications such as de novo protein design model evaluation, no evolutionary multiple sequence alignment information exists. DeepAccNet-Bert includes the Bert embeddings, which are generated with a single sequence without any evolutionary alignments, and it outperformed the DeepAccNet-MSA on the EMA tasks for proteins with no homologous sequence information (Supplementary Fig. 4). DeepAccNet-MSA would be a more robust choice when multiple sequence alignment information is available (Supplementary Table 3). We compared the performance of the DeepAccNet variants on the CASP13 EMA data (76 targets with ~150 decoy models each) to that of the methods that similarly estimate error from a single structure model. These are Ornate (group name 3DCNN)[9], a method from Lamoureux Lab[18], VoroMQA[10], ProQ3[19], ProQ3D, ProQ3D-lDDT[8], and MOD-FOLD7[20]; the former two use 3D convolutions similar to those used in our single residue environment feature calculations. We calculated (i) the Spearman-r of predicted and actual global l-DDT scores per target protein and (ii) area under receiver operator characteristic (ROC) curve for predicting mis-modeled residues per sample (all-atom l-DDT $< 0.6$[21]), which assesses global and local model accuracy estimation, respectively. According to both metrics, DeepAccNet-Standard and DeepAccNet-Bert outperformed the other methods that do not use any evolutionary information; DeepAccNet-MSA also outperformed the other methods that use evolutionary multiple sequence alignment information (Fig. 3b). While this improved performance is very encouraging, it must be noted that our predictions are made after rather than before CASP13 data release so the comparison is not entirely fair: future blind accuracy prediction experiments will be necessary to compare methods on an absolutely even footing. As a step in this direction, we tested performance on structures released from the PDB after our network architecture was finalized that were collected in the CAMEO (Continuous Automated Model EvaluatiOn)[21] experiment between 2/22/2020 to 5/16/2020. We consistently observed that DeepAccNet-Standard and DeepAccNet-Bert improved on other methods that do not use evolutionary information,— namely, VoroMQA[10], QMean3[22], and Equant 2[23] in both global (entire model) and local (per residue) accuracy prediction performance (Supplementary Fig. 5). DeepAccNet-MSA also showed state of the art performance among the methods that use multiple sequence alignments. We could not compare signed residue-pair distance error predictions because this is not predicted by the other methods.

**Guiding search in protein structure refinement using the accuracy predictors.** We next experimented with incorporation of the network accuracy predictions into the Rosetta refinement protocol[5,24], which was one of the top methods tested in CASP13[25]. Rosetta high-resolution refinement starts with a single model, and in a first diversification stage explores the energy landscape around it using a set of sampling operators, and then in a subsequent iterative intensification stage hones in on the lowest energy regions of the space. Search is controlled by an evolutionary algorithm, which maintains a diverse but low energy pool through many iterations/generations. With improvements in the Rosetta energy function in the last several years[26,27], the bottleneck to improving refinement has largely become sampling close to the correct structure. The original protocol utilized model consensus-based accuracy estimations (i.e., regional accuracy estimated as inverse of fluctuation within an ensemble of structures sampled around the input model) to keep search focused in the relevant region of the space—these have the obvious downside of limiting exploration in regions which need to change substantially from the input model but are located in deep false local energy minima.

To guide search, estograms and $C_\beta$ I-DDT scores were predicted and incorporated at every iteration in the Rosetta refinement protocol at three levels (details in Methods). First and most importantly, the estograms were converted to residue–residue interaction potentials with weight for each pair defined by a function of its estogram prediction confidence, and these potentials were added to the Rosetta energy function as restraints to guide sampling. Second, the per-residue $C_\beta$ l-DDT predictions were used to decide which regions to intensively sample or to recombine with other models. Third, global $C_\beta$ l-DDT prediction was used as the objective function during the selection stages of the evolutionary algorithm and to control the model diversity in the pool during iteration.

To benchmark the accuracy prediction guided refinement protocol, 73 protein refinement targets were collected from previous studies[5,24]. The starting structures were generally the best models available from automated structure prediction methods. A separate 7 targets from Park et al.[5,24] were used to tune the restraint parameters and were excluded from the tests below.

We found that network-based accuracy prediction consistently improves refinement across the benchmark examples. In Fig. 4, refinement guided by the accuracy predictions from DeepAccNet-Standard is compared to our previous protocol in which simpler non-deep learning accuracy estimation was used. Refinement of many proteins in the benchmark set was previously quite challenging due to their size[24]; however, with the updated protocol, consistent improvements are observed over the starting models regardless of protein size (Fig. 4a, all-atom l-DDT improve by 10% on average) and over the models produced with our previous unguided search (Fig. 4b; all-atom I-DDT improves by 4% on average). The number of targets with all-atom l-DDT improvements of greater than 10% increases from 27 to 47% using DeepAccNet-Standard to guide refinement. These improvements are quite notable given how challenging the protein structure refinement problem is (comparison to other best predictors on the latest CASP targets is shown in Supplementary Fig. 6); for reference best improvements between successive biannual CASP challenges are typically <2%[25]. Tracing back through the refinement trajectory reveals that the progress in both predicted and actual model quality occurs gradually through the stages and that the two are correlated (Supplementary Fig. 7a). Predictions of more detailed per-residue model quality also agree well with the actual values (Fig. 4e).

We evaluated the practical impact of the improvement in refined model quality using the accuracy predictions by carrying out molecular replacement (MR) trials with experimental diffraction datasets (Fig. 4c). On 41 X-ray datasets from the

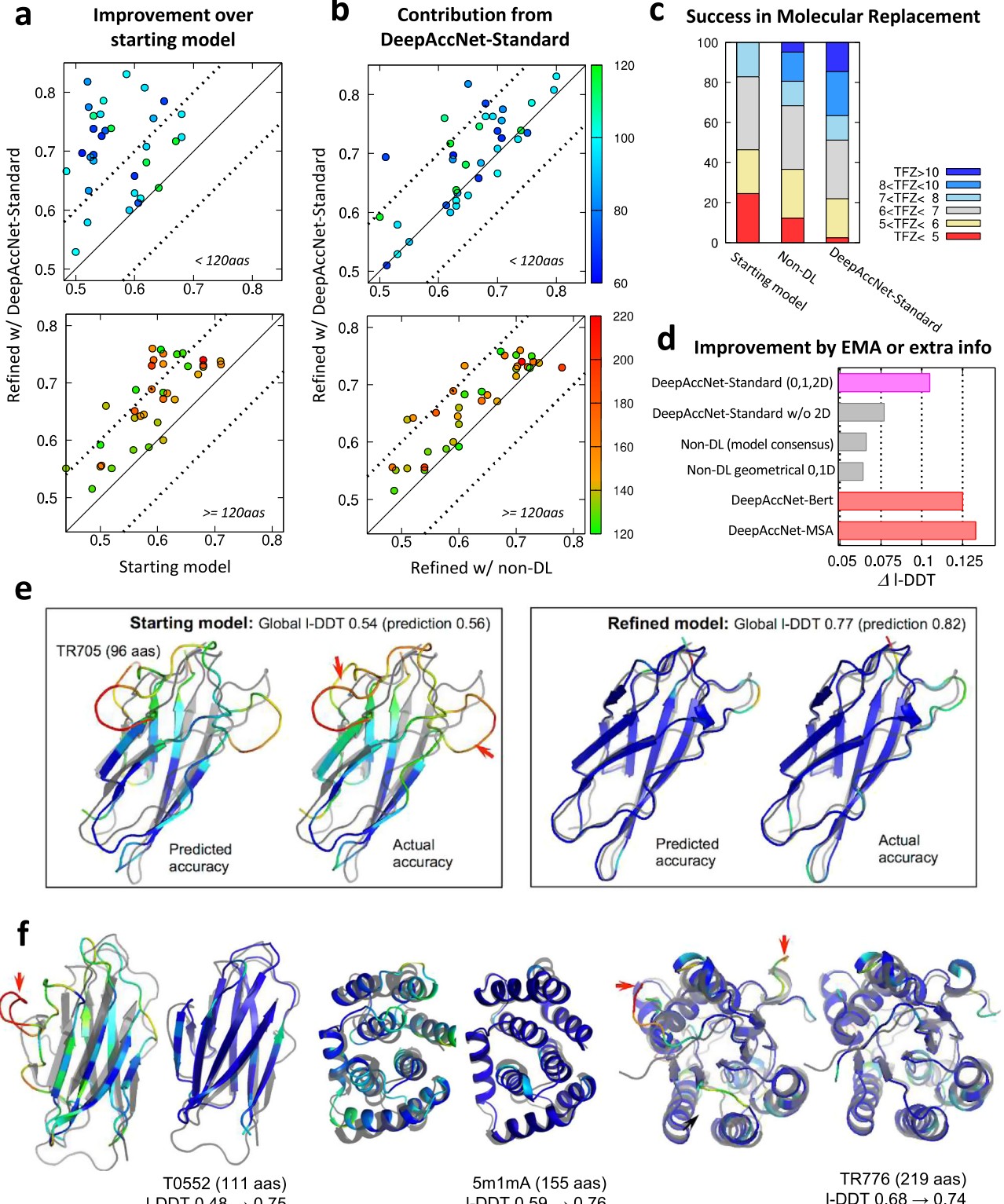

**e** Starting model: Global l-DDT 0.54 (prediction 0.56)

TR705 (96 aas)

Predicted accuracy    Actual accuracy

Refined model: Global l-DDT 0.77 (prediction 0.82)

Predicted accuracy    Actual accuracy

**f**

T0552 (111 aas)
l-DDT 0.48 → 0.75

5m1mA (155 aas)
l-DDT 0.59 → 0.76

TR776 (219 aas)
l-DDT 0.68 → 0.74

benchmark set, the fraction of cases for which robust MR hits were obtained was 0%, 20%, and 37% using prerefined models, models refined by the non-deep learning protocol, and models refined using DeepAccNet-Standard, respectively.

Residue-pair restraints derived from the DeepAccNet estogram predictions were crucial for the successful refinement (Fig. 4d and Supplementary Fig. 8a). When only residue-wise and global accuracy predictions (either from DeepAccNet-Standard or external EMA tool[10]) were utilized for the refinement

calculations, performance did not statistically differ from our previous work (*p*-value > 0.1). When Bert or MSA input was further provided to DeepAccNet (red bars in Fig. 4d), significant increases in model quality was observed for a number of targets (Supplementary Fig. 8b). Final pool model quality analyses (Supplementary Fig. 9) suggest that sampling was improved by those extra inputs (i.e., overall model quality increases) while the single-model selection was generally reasonable across the three different network-based EMAs.

**Fig. 4 Consistent improvement in model structures from refinement runs guided by deep learning based accuracy predictions.** Refinement calculations guided and not guided by network accuracy predictions were carried out on a 73 protein target set[5,24] (see Methods for details). **a** Network guided refinement consistently improves starting model. **b** Network guided refinement trajectories produce larger improvements than unguided refinement trajectories. The accuracy of the refined structure (all-atom l-DDT; y-axis) is compared with that of the starting structure in panel **a**, and with the final refined structure using non-DL-based model consensus accuracy predictions in panel **b**[5]. Top and bottom panels show results for proteins less than 120 residues in length and 120 or more residues in length, respectively. Each point represents a protein target with color indicating the protein size (scale shown at the right side of panel **b**). **c** Molecular replacement experiments on 41 benchmark cases using three different sets of models: (i) starting models, (ii) refined models from the non-deep learning protocol, and (iii) guided by DeepAccNet-Standard. Distributions of TFZ (translation function Z-score) values obtained from Phaser software[37] are reported; TFZ values greater than 8 are considered robust MR solutions. d) Model improvements brought about by utilizing DeepAccNet-Standard (magenta), different EMA methods (gray bars), and other DeepAccNet variants trained with Bert or MSA features (red bars). Average improvements tested on the 73 target set are shown. For the "DeepAccNet-Standard w/o 2D" and "geometrical EMA"[10], residue-pair distance confidences are estimated by the multiplication of residue-wise accuracy following the scheme in our previous work[5,24] (details can be found in Methods; head-to-head comparison shown in Supplementary Fig. 8). **e** Example of predicted versus actual per-residue accuracy prediction. Predicted and actual $C_\beta$ l-DDT values are shown before (left) and after refinement (right) with a color scheme representing local l-DDT from 0.0 (red) to 0.7 (blue). Native structure is overlaid in gray color. Red arrows in the panels highlight major regions that have been improved. **f** Examples of improvements in refined model structures. For each target, starting structures are shown on the left and the refined model on the right. Color scheme is the same as **e**, showing the actual accuracy.

The model accuracy improvements occur across a broad range of protein sizes, starting model qualities, and types of errors. Refinement improved models across different secondary structures to similar extents and corrected secondary structures originally modeled incorrectly, increasing model secondary structure accuracy by almost 10% based on an 8-state definition[28] (Supplementary Fig. 7b, c). As shown in Fig. 4f, improvements involve identification and modifications of erroneous regions when the overall structure is correct (TR776) as well as overall concerted movements when the core part of the model is somewhat inaccurate (5m1mA). The accuracy prediction network promotes this overall improvement in two ways: first, it provides a more accurate estimation of unreliable distance pairs and regions at every iteration of refinement for every model on which sampling can be focused, and second, it provides a means to effectively constrain the search space in the already accurately modeled regions through residue–residue-pair restraints—this is particularly important for refinement of large proteins. The network enables the refinement protocol to adjust how widely to search on a case-by-case basis; this is an advantage over most previous refinement approaches where search has generally been either too conservative or too aggressive[29].

## Discussion

Representations of the input data are critical for the success of deep learning approaches. In the case of proteins, the most complete description is the full Cartesian coordinates of all of the atoms, but these are transformed by rotation and hence not optimal for predicting rotationally invariant quantities such as error metrics. Hence most previous machine learning based accuracy prediction methods have not used the full atomic coordinates[8,10,19]. The previously described Ornate method does use atomic coordinates to predict accuracy, solving the rotation dependence by setting up local reference frames for each residue. As in the Ornate method, DeepAccNet carries out 3D convolutions over atomic coordinates in residue centered frames, but we go beyond Ornate by integrating together this detailed residue information along with additional individual residue and residue–residue level geometric and energetic information by 2D convolutions over the full $N \times N$ residue–residue distance map. DeepAccNet-Bert further employs the sequence embeddings from the ProtBert language model[16], which provides a higher level representation of the amino acid sequence more directly relatable to 3D structures.

Evaluation of performance on CASP13 and CAMEO datasets shows that the DeepAccNet networks make state-of-the-art accuracy predictions, and they were further used to predict signed distance errors for protein structure refinement. Model quality estimations on X-ray crystal structures correlate with resolution, and the network should be useful in identifying errors in experimentally determined structures (Fig. 3c). DeepAccNet performs well on both cryoEM and membrane protein structures, and it could be particularly useful for low-resolution structure determination and modeling of currently unsolved membrane proteins (Fig. 3c). We also anticipate that the network will be useful in evaluating protein design models.

Guiding search using the network predictions improved Rosetta protein structure refinement over a wide range of protein sizes and starting model qualities (Fig. 4). However, there is still considerable room for improvement in the combined method. To more effectively use the information in the accuracy predictions it will be useful to explore sampling strategies which can better utilize the network predictions and more frequent communication between Rosetta modeling and the accuracy prediction network—the network is fast enough to evaluate the accuracy of many models more frequently. Also, we find that DeepAccNet often overestimates the quality of models when those are heavily optimized by the network through our refinement protocol (Supplementary Fig. 7a); adversarial training could help reduce this problem and allow more extensive refinement. It is clear that there is also considerably more to explore in using deep learning to guide refinement. For example, selection of which of the current sampling operators to use in a given situation, and the development of new sampling operators using generative models such as sampling missing regions by inpainting. More generally, reinforcement learning approaches should help identify more sophisticated iterative search strategies.

As a rigorous blind test of both our accuracy prediction and refinement methods, we entered them in the CASP14 structure prediction experiment, and while this manuscript was in the final stages of revision, the independent assessors' results were presented at the CASP14 meeting. Our methods performed quite well, for example the accuracy prediction guided refinement method was the only refinement method at CASP14 able to consistently improve targets greater than 200 amino acids[30]. In the EMA category, both DeepAccNet-Standard and DeepAccNet-MSA were the top single-model methods for global QA (top1 loss), DeepAccNet-MSA was the best single-model method for local QA, and DeepAccNet-Standard was the best single-model local QA method that does not use any coevolutionary information[31,32]. Taken together with the benchmarking experiments described in detail in this paper, these results suggest that

the accuracy prediction and refinement methods are improvements over the previous state of the art.

## Methods

**Data preparation.** Training and test sets for protein model structures (often called decoys) are generated to most resemble starting models of real-case refinement problems. We reasoned that a relevant decoy structure should meet the following conditions: (i) has template(s) not too far or close in sequence space; (ii) does not have strong contacts to other protein chains, (iii) should contain minimal fluctuating (i.e., missing density) regions. To this end, we picked a set of crystal structures from the PISCES server (deposited by May 1, 2018) containing 20,399 PDB entries with maximum sequence redundancy of 40% and minimum resolution of 2.5 Å. We further trimmed the list to 8718 chains by limiting their size to 50–300 residues and requiring that proteins are either monomeric or have minimal interaction with other chains (weaker than 1 kcal/mol per residue in Rosetta energy). HHsearch[33] was used to search for templates; 50 templates with the highest HHsearch probability, sequence identity of at most 40% and sequence coverage of at least 50% are selected for model generation.

Decoy structures are generated using three methods: comparative modeling, native structure perturbation, and deep learning guided folding. Comparative modeling and native structure perturbation are done using RosettaCM[15]. For comparative modeling of each protein chain, we repeated RosettaCM 500 times in total, every time randomly selecting a single template from the list. In order to increase the coverage of decoy structures at mid-to-high accuracy regime for targets lacking templates with GDT-TS > 50, 500 models are further generated providing a single template and 40% trimmed native structure as templates. Sampled decoy set for a protein chain is included in training/test data only if the total number of decoys at medium accuracy (GDT-TS to native ranging from 50 to 90) is larger than 50. Maximum 15 lowest scoring decoys at each GDT-TS bin (ranging from 50 to 90 with bin size 10) are collected, then the rest with lowest energy values are filled so as to make the set contain approximately 90 decoys. Native structures are perturbed to generate high-accuracy decoys. 30 models were generated by RosettaCM either by (i) combining a partial model of a native structure with high-accuracy templates (GDT-TS > 90) or (ii) inserting fragments at random positions of the native structure. Deep learning guided folding is done using trRosetta[1]. For each protein, five subsampled multiple sequence alignments (MSAs) are generated with various depths (i.e., number of sequences in MSA) ranging from 1 to maximum available. The standard trRosetta modeling is run 45 times for each of the subsampled MSAs. The final decoy set collected, consisting of about 150 structures (90 from comparative modeling, 30 from native perturbation, and 30 from deep learning guided folding) per each of 7307 protein chains (6749, 280, 278 for training, validation and test datasets), are thoroughly relaxed by Rosetta dual-relax[34] prior to the usage. The distribution of the starting $C_\beta$ l-DDT values of the test proteins are shown in Supplementary Fig. 10.

**Model architectures and input features.** In our framework, convolution operations are performed in several dimensions, and different classes of features come in at different entry points of the network (Fig. 1). Here, we briefly describe the network architecture as well as classes of features. More detailed descriptions about the features and model parameters are listed in Supplementary Tables 4 and 5.

The first set of input features to the network are voxelized Cartesian coordinates of atoms per residue, generated in a manner similar to Ornate[9]. Voxelization is performed individually for every residue in the corresponding local coordinate frame defined by backbone N, Cα, and C atoms. Such representation is translationally and rotationally invariant because projections onto local frames are independent of the global position of the protein structure in 3D space. The second set of inputs are per-residue 1D features (e.g., amino acid sequence and properties, backbone angles, Rosetta intraresidue energy terms, and secondary structures) and per-residue-pair 2D features (e.g., residue–residue distances and orientations, Rosetta inter-residue energy terms, inter-residue distance predictions from the trRosetta network[1], and the ProtBert-BFD100 embeddings[16]).

In the first part of the neural network, the voxelized atomic coordinates go through a series of 3D-convolution layers whose parameters are shared across residues. The resulting output tensor is flattened so that it becomes a 1D vector per residue, which is concatenated to other 1D features. The second part of the network matches the dimensionality of the features and performs a series of 2D convolution operations. Let us now denote that there are $n$ residues, $f_1$ 1D features, and $f_2$ 2D features. Then, the input matrix of the 1D features $M_1$ has the shape of $n$ by $f_1$, and the input matrix of the 2D features $M_2$ has the shape of $n$ by $n$ by $f_2$. We tile $M_1$ in the first and second axis of $M_2$, concatenating them to produce a feature matrix of size $n$ by $n$ by $2f_1 + f_2$. The third axis of the resulting matrix represents vectors of size $2f_1 + f_2$, which contain the 2D features and 1D features of $i$-th and $j$-th residues. This data representation allows us to convolve over both backbone chain and pairwise interactions.

The concatenated feature matrix goes through a residual network with 20 residual blocks, with cycling dilation rates of 1, 2, 4, and 8 (see Supplementary Table 5). Then, the network branches off to two arms of four residual blocks. These arms separately predict distributions of $C_\beta$ distance errors for all pairs of residues (referred to as estograms) and whether a particular residue pair is within 15 Å in a

corresponding native structure (referred to as masks). Estograms are defined over categorical distributions with 15 binned distance ranges; the boundary of bins are at −20.0, −15.0, −10.0, −4.0, −2.0, −1.0, −0.5, 0.5, 1.0, 2.0, 4.0, 10.0, 15.0, and 20.0 Å.

In the standard calculation of a $C_\beta$ l-DDT score of $i$-th residue of a model structure, all pairs of $C_\beta$ atoms that include the $i$-th residue and less than 15 Å in a reference structure are examined. 0.5, 1.0, 2.0, and 4.0 Å cutoffs are used to determine the fractions of preserved $C_\beta$ distances across the set of pairs. The final $C_\beta$ l-DDT score is calculated by computing the arithmetic mean of all fractional values[13].

In our setup, we obviously do not have access to reference native structures. Instead, a $C_\beta$ l-DDT score of $i$-th residue is predicted by combining the probabilistic predictions of estograms and masks as follows:

$$\text{per\_residue\_LDDT} = 0.25 * (\bar{p}_0 + \bar{p}_1 + \bar{p}_2 + \bar{p}_3)/\bar{p}_4 \quad (1)$$

$\bar{p}_0$ is the mean of probability that the magnitudes of $C_\beta$ distance errors are less than 0.5 Å, across all residue pairs that have $i$-th residue involved and predicted to be less than 15 Å in its corresponding native structure. The former $C_\beta$ distance errors are obtained from estogram predictions and the latter native distance information are directly obtained from mask predictions. $\bar{p}_1 \ldots \bar{p}_3$ are similar quantities with different cutoffs for errors; 1.0, 2.0, and 4.0 Å, respectively. $\bar{p}_4$ is the mean probability that native distance is within 15 Å and it is again directly obtained from mask predictions.

The network was trained to minimize categorical cross-entropy between true and predicted estograms and masks. Additionally, as noted, we calculated $C_\beta$ l-DDT scores based on estograms and masks, and we used a small amount of mean squared loss between predicted and true scores as an auxiliary loss. The following weights on the three loss terms are used.

$$\text{global\_loss} = \text{estogram\_loss} + 10.0 * \text{LDDT\_loss} + 0.25 * \text{mask\_loss} \quad (2)$$

The weights are tuned so that the highest loss generally comes from estogram_loss since estograms are the richest source of information for the downstream refinement tasks. At each step of training, we selected a single decoy from decoy sets of a randomly chosen training protein without replacement. The decoy sets include native structures, in which case the target estograms ask networks to not modify any distance pairs. An epoch consists of a full cycle through training proteins, and the training processes usually converge after 100 epochs. Our predictions are generated by taking an ensemble of four models in the same training trajectory with best validation performance. We used an ADAM optimizer with a learning rate of 0.0005 and decay rate of 0.98 per epoch. Training and evaluation of the networks was performed on RTX2080 GPUs.

**Analyzing the importance of features.** Feature importance analysis was conducted to understand and quantify the contributions from different classes of features to accurately predicting accuracy of model structures. To do this, we combined each feature class with a distance map one at a time during training (or removed them in one particular case) and analyzed loss of predictions on a held-out test protein set. In addition to the DeepAccNet-Standard, -Bert, and -MSA, we trained eight types of networks: (i) distance map only, (ii) distance with local atomic environments scanned with 3D convolution, (iii) distance with Bert embeddings, (iv) ii and iii combined, (v) distance with Rosetta energy terms, (vi) distance with amino acid identities and their properties, (vii) distance with secondary structure information, and (iv) distance with backbone angles and residue–residue orientations. For each network, we took an ensemble of four models with best validation performance from the same trajectory in order to reduce noise.

We are aware that more sophisticated feature attribution methods for deep networks exist[35]; however, these methods attribute importance scores to features per output per sample. Since we have approximately a quarter million outputs and near million inputs with a typical 150 residue protein, these methods were not computationally feasible and tractable to analyze.

**Comparing with other model accuracy estimation methods.** For the CASP13 datasets, we downloaded submissions of QA139_2 (ProQ3D), QA360_2 (ProQ3D-lDDT[8]), QA187_2 (ProQ3[19]), QA067_2 (LamoureuxLab[18]), QA030_2 (VoroMQA-B[10]), QA275_2 (MODFOLD7), QA359_2 (Ornate, group name 3DCNN[9]) for the accuracy estimation category. The former five methods submitted their predictions for 76 common targets, whereas the last method, Ornate, only submitted for 55 targets. Thus, we decided to analyze predictions on the 76 common targets from all methods except for Ornate, which was only evaluated on 55 targets. An evaluation was performed in two metrics; (i) Spearman-r of predicted and true global quality scores across decoys of each target, and (ii) area under ROC curve for predicting mis-modeled residues of each sample (all-atom l-DDT < 0.6). The latter metric is one of the official CAMEO metrics for local accuracy evaluation. Samples whose residues are all below or above 0.6 all-atom l-DDT are omitted. For assessing the performance of methods other than ours, their submitted estimations of global quality scores were evaluated against the true all-atom global l-DDT scores. For DeepAccNet, we use mean $C_\beta$ l-DDT as global quality score.

For the CAMEO datasets, we downloaded the QA datasets registered between 2/22/2020 to 5/16/2020. This corresponds to 206 targets with ~10 modeled

structures on average. We downloaded submissions of "Baseline potential", EQuant 2, ModFOLD4, ModFOLD6, ModFOLD7_LDDT, ProQ2, ProQ3, ProQ3D, ProQ3D_LDDT, QMEAN3, QMEANDisco3, VoroMQA_sw5, and VoroMQA_v2. Some methods did not submit their predictions for all samples, and those missing predictions are omitted from the analysis.

**Visualizing predictions**. Figure 2 visualizes true and predicted estograms per pair of residues. The images are generated by calculating the expected values of estograms by taking weighted sums of central error values from all bins. For the two bins that encode for errors larger than 20.0 Å and smaller than −20.0 Å, we define the central distance at their boundaries of 20.0 Å and −20.0 Å.

**Native structure dataset**. Native structures that were not used for model training and validation, monomeric, larger than 40 residues, and smaller than 300 residues for the X-ray and NMR structures, and smaller than 600 residues for EM structures were downloaded from the PDB. For Fig. 3c, samples with a resolution larger than 4 and 5 Å are ignored for the X-ray and EM structures, respectively. The histograms in Fig. 3d are using all samples without any resolution cutoff. In total, 23,672 X-ray structures, 88 EM structures, and 2154 NMR structures are in the histograms (Fig. 3d). For NMR structures, regions highly varying across the models were trimmed. Structures were discarded if the number of remaining residues after trimming was less than 40 residues or half of the original chain length.

**CASP14**. For the EMA category, DeepAccNet-standard was registered as "BAKER-experimental (group 403)", and DeepAccNet-MSA was registered as "BAKER-ROSETTASERVER (group 209)"[31]. For the refinement category, our protocol was registered as "BAKER (group 473)"[30].

**Dataset for refinement runs**. We took 73 proteins and their starting models from our previous work[5] with a few modifications as described below. Of the entire 84 targets used in our previous work, seven small-sized targets (4vz5A, 5azxA, 5ghaE, 5i2qA, 5xgaA, TR569, and T0743) are excluded from the benchmark set and were used for restraint parameter search. Eight additional targets (2n12A, 4idiA, 4z3uA, 5aozA, 5fidA, T0540, TR696, and TR857) are excluded after more careful visual inspections as those had potential issues in their native structures (e.g., having contacts with ligands or other chains in crystal structures). Four new targets were added from previous CASP refinement categories that were not included in the original set (TR747, TR750, TR776, and TR884). Model accuracy is evaluated on a subset of ordered residues by trimming less confident residues according to the CASP standard evaluation criteria[25].

**Refinement protocol**. Refinement protocol tested in this work inherits the framework from previous study[5]. The overall architecture consists of two stages (Fig. 1b): first initial model diversification stage, followed by iterative model intensification stages where a pool of structures is maintained during optimization by an evolutionary algorithm. At the diversification stage, following accuracy estimation of the single starting model, 2000 of independent Rosetta modeling are attempted using RosettaCM[15]. In the iterative annealing stage, series of accuracy estimation, new structure generation, and pool selection steps are repeated iteratively. At each iteration, 10 model structures are selected from the current pool, then individual accuracy predictions are made for each of 10 structures in order to guide the generation of 12 new model structures starting from each (total 120). New pool with size of 50 is selected among 50 previous pool members plus 120 newly generated ones with criteria of (i) the highest global $C_\beta$ l-DDT estimated and (ii) model diversity within the pool. This process is repeated for 50 iterations. At every fifth iteration, a recombination iteration is called instead of a regular iteration where model structures are recombined with another member in the pool according to the residue $C_\beta$ l-DDT values predicted by the network (see below).

For modeling of a single structure at both diversification and intensification stages, first unreliable regions in the structure are estimated from accuracy prediction (see below). Structural information is removed in those regions and fully reconstructed from scratch. Fragment insertions are carried out in a coarse-grained broken-chain representation of the structure[15] focusing more on unreliable regions (five times more frequently with respect to the rest part), followed by repeated side-chain rebuilding and minimization[34] in all-atom representation. Both coarse-grained and all-atom stage modeling are guided by distance restraints derived from accuracy predictions in addition to Rosetta energy. Details of unreliable region predictions, recombination iteration, and restraints are reported in the following sections.

**Unreliable region prediction**. Accuracy values predicted from the network are used to identify unreliable regions. We noticed that the l-DDT metric has a preference for helical regions (as local contacts are almost always correct). To fix this systematic bias, we exclude short sequence separation contacts in the contact mask that are within sequence separation of 11 to get corrected residue $C_\beta$ l-DDT values. Then these values are smoothed through a 9-residue-window uniform weight kernel. The residues at the lowest accuracy are determined as unreliable regions. Two definitions of regions are made: in static definition, the accuracy

threshold is varied until the fraction of unreliable regions lies between 10 and 20% of the entire structure. In *dynamic* definition, this range is defined as a function of predicted global accuracy (i.e., average residue-wise corrected accuracy): from $f_{dyn}$ to $f_{dyn}$ +10% with $f_{dyn} = 20 + 20*(0.55 − Q)/30$, where $Q$ refers to predicted global accuracy. $f_{dyn}$ is capped between 20 and 40%. In the diversification stage, one thousand models were generated for each definition of unreliable regions. Static definition is applied throughout the iterative stage.

**Restraints**. We classified residue pairs in three confidence levels: high confidence, moderate confidence, and nonpreserving. Highly or moderately confident residue pairs stand for those whose distance should be fixed from the reference structure (i.e., starting structure) at different strengths; nonpreserving pairs refer to the rest which can freely deviate.

Confident pairs are collected if $C_\beta$–$C_\beta$ distance are not greater than 20 Å and whose "probability with absolute estimated error ≤1 Å", shortly $P_{cen}$, is above a certain threshold (e.g., 0.7). For those pairs, bounded functions are applied at coarse-grained modeling stage, and sum of sigmoid functions at all-atom modeling stage, minima centering at the original distance $d_0$ for both cases:

Bounded function:

$$f(d) = \frac{(d − (d_0 + tol + s))}{s} + 1 \quad \text{for } d > d_0 + tol + s$$
$$= \left(\frac{d − (d_0 + tol)}{s}\right)^2 \quad \text{for } d_0 + tol \leq d \leq d_0 + tol + s$$
$$= 0 \quad \text{for } |d − d_0| < tol \tag{3}$$
$$= \left(\frac{d − (d_0 − tol)}{s}\right)^2 \quad \text{for } d_0 − tol − s \leq d \leq d_0 − tol$$
$$= \frac{(d − (d_0 − tol − s))}{s} + 1 \quad \text{for } d < d_0 − tol − s$$

Sum of sigmoid function:

$$f(d) = w_{fa} * \left[\frac{−1}{1 + \exp(−5.0*(d − d_0 + tol)/s)} + \frac{1}{(1 + \exp(−5.0*(d − d_0 − tol)/s)} + 1\right] \tag{4}$$

where s and tol stand for width and tolerance of the functions. Thresholds in $P_{cen}$ values for highly confident pairs, $P_{high}$, and moderately confident pairs, $P_{moderate}$, are set at 0.8 and 0.7, with $(s, tol) = (1.0, 1.0)$ and $(2.0, 2.0)$, respectively, by analyzing the network test results shown in Supplementary Fig. 11. Restraint weight at all-atom stage modeling, $w_{fa}$, is set as 1.0. We noticed iterative refinement with these empirically determined parameters ($w_{fa}$, {$P_{high}$, $P_{moderate}$}) brought too conservative changes. We therefore ran another iterative refinement with a more aggressive parameter set (0.2, {0.8, 0.9}) and chose the trajectory from whichever sampled a higher predicted global $C_\beta$ l-DDT.

For the rest nonpreserving $C_\beta$–$C_\beta$ pairs whose input distances are shorter than 40 Å, error probability profiles (estograms) are converted into distance potentials by subtracting error bins from the original distances $d_0$ and taking log odds to convert probability into energy units. Instead of applying raw probabilities from the network, corrections are made against background probability collected from the statistics of the network's predictions over 20,000 decoy structures in the training set conditioning on sequence separation, original distances $d_0$, and predicted global model quality. The potential was applied in full form interpolated by spline function at the initial diversification stage, and was replaced by a simpler functional form in subsequent iterative process for efficiency:

$$f(d) = (d − 9) + 1 \quad \text{for } d > 9\,\text{Å} \tag{5}$$

$$f(d) = (d − 8)^2 \quad \text{for } 8 \leq d \leq 9\,\text{Å} \tag{6}$$

$$f(d) = 0 \quad \text{for } d < 8\,\text{Å} \tag{7}$$

for those pairs predicted from estogram as contacting within 10 Å. Contacts are predicted when $P_{contact} > 0.8$, with $P_{contact} = \text{sum}(P_i)$ over i whose $d_0 + e_i < 10$ Å and $P_i$ stands for probability in estogram at bin i.

**Recombination Iteration**. At the recombination iteration, instead of running RosettaCM as the sampling operator, model structures are directly generated by recombining the coordinates from two models according to the predicted residue $C_\beta$ l-DDT profiles by the network. For a "seed" member, 4 "partners" are identified among the remaining 49 members in the pool that have the most complementarity to the seed in the predicted residue $C_\beta$ l-DDT profiles. All the members in the pool are recombined individually with their 4 partners, resulting in a total 200 new structural models. For each seed-partner combination, first, "complementary regions" are identified where the seed is inferior to the partner in terms of predicted $C_\beta$ l-DDT, then coordinates at the regions are substituted to those from the partner. Multiple discontinuous regions are allowed but the total coverage is restricted to a range between 20 and 50% of total residues. Next, Rosetta FastRelax[34] is run by imposing residue-pair restraints from estograms brought from either the partner or the seed interpolated into pair potentials (see above). Restraints from the partner are taken if any residue in the pair is included in complementary regions, and from

the seed for the rest pairs. Recombination iterations are called at every five iterations to prevent overconvergence in the pool.

**Final model selection**. A model with the highest predicted global $C_\beta$ l-DDT is selected among 50 final pool members. Then a pool of structures similar to this structure (S-score[36] > 0.8) are collected from the entire iterative refinement trajectory, structurally averaged, and regularized in model geometry by running dual-relax[34] with strong backbone coordinate restraints with a harmonic constant of 10 kcal/mol Å², which was the identical post-processing procedure in our previous work[5]. The final model refers to this structurally averaged and subsequently regularized structure. Structural averaging adds 1% all-atom l-DDT gain on average.

**Testing other EMA methods in the refinement protocol**. To test the refinement coupled with an external non-DL geometrical EMA, VoroMQA[10] version 1.21 was downloaded and integrated into our refinement protocol script substituting DeepAccNet for global model ranking and unreliable region prediction. Because VoroMQA does not provide any residue-pair estimations, confidence in the distance between residue $i,j$ (denoted as $P_{ij}$) was estimated by the logic used in our previous work[5,24]. Here, $P_{ij} = P_i * P_j$ where $P_i = \exp(-\lambda/L_i)$ and $L_i$ is the residue-wise accuracy from VoroMQA; $\lambda$ was set to 1.4, which gave the most similar distribution in weights as what was found in our previous work. Then $P_{ij}$ was divided by the highest 70 percentile value capping the maximum value at 1.0. Residue-pair restraints were applied at these per-pair weights with the identical functional forms described above. The same logic was applied to the refinement protocol using "DeepAccNet w/o 2D"; here $\lambda = 2.0$ was used.

**Molecular replacement (MR)**. Of a total 50 targets native structures of which were determined by X-ray crystallography in the benchmark set, 41 are tested for MR. Nine targets are excluded as their crystal structures contained other proteins or domains with significant compositions (>50%). Phaser[37] in the Phenix suite version 1.18rc2-3793 is applied with MR_AUTO mode. Terminal residues are trimmed from model structures prior to MR if they do not directly interact with the rest of residues. B-factors are estimated by taking residue-wise DeepAccNet predictions: first, $u_i$, the position error at residue i (in Å), is estimated by using a formula: $u_i = 1.5*\exp[4*(0.7 - lddt_{predicted,i})]$, where parameters were prefit to training set decoy structures. Then B-factor at residue $i$ is calculated as $8\pi^2 u_i^2/3$.

## Data availability
Decoy structures generated for the training of the DeepAccNet models and their raw predictions on the held-out test, CASP13, and CAMEO set are available at the github repository https://github.com/hiranumn/DeepAccNet. Other relevant data are available upon request.

## Code availability
Code and accompanying scripts for the model accuracy predictors (DeepAccNet-Standard, DeepAccNet-MSA, and DeepAccNet-Bert) are implemented and made available at https://github.com/hiranumn/DeepAccNet.

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

## Acknowledgements

We would like to thank Su-In Lee, Frank DiMaio, Sanaa Mansoor, Doug Tischer, and Alex Yuan for helpful discussions. This work is supported by Eric and Wendy Schmidt by recommendation of the Schmidt Futures program (N.H. and H.P.), Washington Research Foundation (M.B.), NIAID Federal Contract # HHSN272201700059C (I.A.), The Open Philanthropy Project Improving Protein Design Fund (J.D.), and Howard Hughes Medical Institute and The Audacious Project at the Institute for Protein Design (D.B.).

## Author contributions

N.H., H.P., and D.B. designed research; N.H., I.A., M.B., and J.D. contributed in developing the deep learning networks; N.H. analyzed the network results; H.P. contributed to the application of the network on the refinement process; N.H., H.P., and D.B. wrote the manuscript.

## Competing interests

The authors declare no competing interests.
