## [Peer Review File · Nature Communications]

REVIEWER COMMENTS

Reviewer #1 (Remarks to the Author):

In this paper the authors describe a novel method DeepAccNet to predict the quality of a protein model. It performs as well (if not better than) existing QA methods. However, in contrast to earlier methods it also seems to be useful for protein refinement, largely increase the usefulness of these methods.

It will be exciting to see how it performs in CASP14

The paper is well written and the tests appear to be done in an appropriate way (using reasonable methods for cross-validation).

Reviewer #2 (Remarks to the Author):

This manuscript presents a new deep learning based protein model accuracy predictor called DeepAccNet, which, when incorporated into the Rosetta refinement protocol, results in improved structure refinement. The prediction of inter-residue error histograms to guide structure refinement is novel and interesting. The use of residue-level IDDT prediction in refinement is relatively new. DeepAccNet has been shown to deliver better accuracy estimation performance compared to several other single-model-based methods in CASP and CAMEO datasets. Furthermore, structure refinement guided by deep learning based accuracy estimation has been shown to deliver improved refinement performance in a benchmark set of 73 refinement targets. DeepAccNet is available as an open source tool useful for the community.

Major Comments:

1.) My biggest concern is the lack of generalizability of DeepAccNet for protein model accuracy estimation task in its current form. The authors did not use the CASP13 dataset as it is for testing, but rather preprocessed all models using dual space relaxation in Rosetta before model quality estimation (this is possibly the case for the CAMEO dataset as well, although it is entirely not clear from the manuscript). Employing dual space relaxation might have altered the energetics of the original models to make them more "Rosetta friendly", thereby potentially losing the modeling diversity of the original dataset. This, in my opinion, defeats the entire purpose of model accuracy estimation, in which it is expected that there may be a great deal of diversity in modeling procedures (e.g., CASP EMA category). The issue is closely tied to the nature of the training data, which uses various Rosetta protocols for decoy generation. As such, it is not clear how useful DeepAccNet is beyond the realm of Rosetta. Can the authors demonstrate the generalizability of DeepAccNet without preprocessing the models via dual space relaxation in Rosetta for CASP13 and CAMEO datasets?

2.) The authors did not use multiple sequence alignment information in DeepAccNet development based on the assumption that all homology/co-evolutionary information may have been already used in the modeling stage prior to refinement and/or evolutionary information may not be available for de novo protein design model evaluation. This is not necessarily a reasonable assumption to make in protein structure prediction applications without prior knowledge of the modeling protocol used. For example, if the starting structure for refinement is generated via pure physics-based MD simulation without using any homology or co-evolutionary information, then the use of multiple sequence alignment information may turn out to be critical for the prediction of inter-residue error histograms in order to guide refinement. Even when using homology or co-evolutionary information, a certain modeling protocol may not have fully exploited the available information in the form of restraints due to a premature convergence during its optimization. In such a situation, evolutionary information derived from multiple sequence alignment may be highly effective for model accuracy estimation and refinement. That is, depending on the nature and efficacy of the modeling protocol, the use of multiple sequence alignment may turn out to be very

useful. Therefore, I suggest that the authors include multiple sequence alignment information in DeepAccNet and see if it can further improve the accuracy estimation performance by comparing it head-to-head with the state-of-the-art model quality estimation methods that incorporate evolutionary information. The authors can then use this evolutionary-powered DeepAccNet for evaluating structure refinement performance in diverse protein modeling scenarios by generating starting structure without utilizing homology or co-evolutionary information and/or partially using evolutionary information.

3.) The choice of C_{β} IDDT threshold of 0.6 for evaluating the performance of predicting mis-modeled residues seems arbitrary. Also, samples whose residues are all below or above 0.6 C_{β} IDDT are omitted. What is the basis of the choice of the threshold 0.6?

4.) Where are the predicted unreliable regions located? Is there any relationship between the secondary structures and the predicted unreliable regions?

5.) Can the authors report the accuracy of the final models (also see comment 2 above) selected with the highest predicted global IDDT (i.e., without any post-processing and/or structure averaging)?

Minor Comments:

6.) Following the local bias removal process during unreliable region prediction by excluding contacts in the contact mask that are within sequence separation of 11, it might be interesting to exclude contacts that are within sequence separation of 23 and focus only on long-range contacts.

7.) Line 274: typo? “..in then..”

Reviewer #3 (Remarks to the Author):

This paper describes a deep learning approach to the problem of estimating the quality of protein models. The authors also explore how these estimates can be used to enhance the sampling of conformations during refinement using Rosetta. The paper is not that clearly written in several important areas, and I found it very hard to work out exactly how exactly the input features were derived, even after looking at the code.

The proposed approach is that of using 3-D convolutional units in a deep neural net to analyse the locations of atoms in the model being evaluated. This is not particularly novel, as several groups have already developed methods based on this same idea, and these papers have been cited in the text. The results presented in the paper are, however, good, certainly amongst the top-tier of methods in this general area, but these improvements are probably more down to using larger and/or better quality data sets, along with other improvements in the mechanics of how the networks were trained, than any methodological advances.

As already mentioned, the overall results look good. I was a little unclear on whether there was any possibility of testing set contamination, however. CASP13 results feature prominently in the results shown, but I was surprised that the authors picked a cutoff date after the end of the CASP13 experiment for compiling their training data. Why not set the cutoff date just before the start of the original experiment? Surely that would be a more realistic comparator with other methods used in that experiment? I was also not sure whether the authors specifically excluded the CASP13 target structures that were released before Nov 2018 from their training data - certainly I can't immediately see any mention of them checking for this using something like ECOD, or did I just miss it?

The authors go to great lengths to separate methods which use multiple sequence alignments from those that don't, but I'm really not sold on their excuse for this. Some proteins won't have deep multiple sequence alignments, that's true. However, very few sequences are orphans given the current databank sizes, particularly with the availability of metagenomic sequences, which the Baker lab has already used in the past to good effect. They argue that protein design targets

would not have alignments available, which is also true, but it's not clear to me how useful this study would be for protein design. The main subject of the study is clearly protein structure prediction and modelling, not protein design. Certainly for protein modelling and refinement, there is no reason whatsoever to exclude multiple sequence alignments from use. The problem is that when MSA based methods are included in the benchmark, the results in supplementary material (Fig. S3) show that DeepAccNet is not appreciably better than some older methods which have traditionally done well at CASP e.g. ModFOLD. It might be unfair to say this, but I have to wonder if the true reason behind excluding those methods from the main text is to try to upsell the DeepAccNet results somewhat. Certainly, I believe that Figure S3 should be included in the main text (Figure 2 can easily be dumped to supplementary material to make space if needed).

The results on refinement seem reasonable enough. However, it's a shame that they did not test against another model quality method rather than just testing with and without their own method. I can see this would have been difficult to implement, though. I'm a bit concerned that the authors may not have excluded the effects of homology from their analysis here. Looking at Figure 4, a refinement jump from a GDT-HA score of 0.44 to 0.72 is massive – but I have to wonder if that target T0552 does actually have homologs in the training data. It's very hard for me to work this out given the numerous different splits and filter criteria the authors mention as they go along. Certainly, I would ask for firm evidence whether any of the cherry-picked examples in Figure 4 overlap in any way with the data used to train DeepAccNet or which has been used for validation/parameter tuning.

Finally, my main gripe. Feature selection is a key component of machine learning studies – homing in on only useful features makes ML models more robust and more importantly produces a more scientifically useful result. If there is smoke coming out of the engine of my car, I don't need to check the tire pressure or the color of the paintwork to work out where the problem is that needs fixing. In a good ML study, feature importance analysis is carried out quite early on, and only the useful features retained in the final study. Here, however, feature importance analysis has been left until the writing of the paper, and that is very poor engineering. The problem for me shows clearly in Figure 3A. What that appears to be telling me is that apart from the distances derived from the starting model, the only other features that are needed are the 3-D convolutions. The difference between plot i) and iii) looks close to zero to me. So basically, all the other stuff that the authors calculate and feed into the network simply aren't worth the effort of calculating them. So why isn't DeepAccNet just defined as being the model that takes the CB distances + 3D convolution? Adding the whole PyRosetta package as a code dependency, for example, is a huge complication (admittedly probably not for the Baker lab) when it's clearly not even needed. What's the point in all the extra wasted effort? These extra features make the work unnecessarily complicated, much harder to use and much harder for others to replicate and build upon. Unless the authors can show statistically that adding those extra features improves on results over the baseline, I strongly believe they should redefine their default approach to be that shown in Fig. 3Aiii – of course in that case the issue might remain as to how much novelty is left in the method, and whether it might be better dealt with in a more targeted journal e.g. Bioinformatics. However, I leave that for the Editors to decide.

A final small point I would raise about the code & submission checklist. The authors state that the existing MIT license, which is fine, is "subject to change". I'm sorry, but I don't think that is acceptable. The authors need to decide under what license they are releasing the code needed to replicate the results of their study and stick to it. They are free to fork the code in the future, as is anybody else, but they must leave the current version of the code available as-is and under the same open source license it had when reviewed.

Thank you for considering our manuscript "Improved protein structure refinement guided by deep learning based accuracy estimation" for review by Nature Communications and for allowing us to submit the point-by-point response and incorporate the comments that we have received into a revision of the manuscript.

This is a resubmission of manuscript **NCOMMS-20-32430**. We would like to thank all reviewers for their careful consideration of this manuscript and their many suggestions for improvement in this round. We worked on addressing the reviewers' comments by adding additional experiments and their analysis to our manuscript. Below, we provide our answers comment by comment.

Reviewer #1 (Remarks to the Author):

In this paper the authors describe a novel method DeepAccNet to predict the quality of a protein model. It performs as well (if not better than) existing QA methods. However, in contrast to earlier methods it also seems to be useful for protein refinement, largely increases the usefulness of these methods.

It will be exciting to see how it performs in CASP14

The paper is well written and the tests appear to be done in an appropriate way (using reasonable methods for cross-validation).

We are glad to hear this. We agree that the strength of our method is not only its state of the art performance on typical quality estimation tasks but also its ability to be directly coupled with the refinement process. We hope that our new Figure 4D puts further emphasis on this point.

Reviewer #2 (Remarks to the Author):

This manuscript presents a new deep learning based protein model accuracy predictor called DeepAccNet, which, when incorporated into the Rosetta refinement protocol, results in improved structure refinement. The prediction of inter-residue error histograms to guide structure refinement is novel and interesting. The use of residue-level IDDT prediction in refinement is relatively new. DeepAccNet has been shown to deliver better accuracy estimation performance compared to several other single-model-based methods in CASP and CAMEO datasets. Furthermore, structure refinement guided by deep learning based accuracy estimation has been shown to deliver improved refinement performance in a benchmark set of 73 refinement targets. DeepAccNet is available as an open source tool useful for the community.

Major Comments:

1.) My biggest concern is the lack of generalizability of DeepAccNet for protein model accuracy estimation task in its current form. The authors did not use the CASP13 dataset as it is for testing, but rather preprocessed all models using dual space relaxation in Rosetta before model quality estimation (this is possibly the case for the CAMEO dataset as well, although it is entirely not clear from the manuscript). Employing dual space relaxation might have altered the energetics of the original models to make them more “Rosetta friendly”, thereby potentially losing the modeling diversity of the original dataset. This, in my opinion, defeats the entire purpose of model accuracy estimation, in which it is expected that there may be a great deal of diversity in modeling procedures (e.g., CASP EMA category). The issue is closely tied to the nature of the training data, which uses various Rosetta protocols for decoy generation. As such, it is not clear how useful DeepAccNet is beyond the realm of Rosetta. Can the authors demonstrate the generalizability of DeepAccNet without preprocessing the models via dual space relaxation in Rosetta for CASP13 and CAMEO datasets?

Thank you for this comment. The statement about dual relaxation was actually unintentionally left over from a previous version of this paper. Our analysis, both on the CAMEO and CASP13 datasets, was done without dual relaxation of modeled structures, which means the network can be used without preprocessing the models. We removed the dual relaxation statement accordingly from our manuscript.

2.) The authors did not use multiple sequence alignment information in DeepAccNet development based on the assumption that all homology/co-evolutionary information may have been already used in the modeling stage prior to refinement and/or evolutionary information may not be available for de novo protein design model evaluation. This is not necessarily a reasonable assumption to make in protein structure prediction applications without prior knowledge of the modeling protocol used. For example, if the starting structure for refinement is generated via pure physics-based MD simulation without using any homology or co-evolutionary information, then the use of multiple sequence alignment information may turn out to be critical for the prediction of inter-residue error histograms in order to guide refinement. Even when using homology or co-evolutionary information, a certain modeling protocol may not have fully exploited the available information in the form of restraints due to a premature convergence during its optimization. In such a situation, evolutionary information derived from multiple sequence alignment may be highly effective for model accuracy estimation and refinement. That is, depending on the nature and efficacy of the modeling protocol, the use of multiple sequence alignment may turn out to be very useful. Therefore, I suggest that the authors include multiple sequence alignment information in DeepAccNet and see if it can further improve

the accuracy estimation performance by comparing it head-to-head with the state-of-the-art model quality estimation methods that incorporate evolutionary information. The authors can then use this evolutionary-powered DeepAccNet for evaluating structure refinement performance in diverse protein modeling scenarios by generating starting structure without utilizing homology or co-evolutionary information and/or partially using evolutionary information.

Thank you for the great suggestion. We additionally trained two new models that include (i) MSA information in the form of trRosetta prediction and (ii) the protein sequence embedding from the ProtBert-BFD100 model (Bert). The performance of these models are described throughout the paper (mostly in Figure 3). Overall these two features significantly improve the network performance. We also ran refinements on the same protein set guided by the newly trained DeepAccNet versions incorporating Bert or MSA as inputs, results of which are reported in Figure 4D and Figure S5 in the revised manuscript.

3.) The choice of C_{β} IDDT threshold of 0.6 for evaluating the performance of predicting mis-modeled residues seems arbitrary. Also, samples whose residues are all below or above 0.6 C_{β} I-DDT are omitted. What is the basis of the choice of the threshold 0.6?

The choice of the threshold 0.6 for ROC and PR is one of the metrics officially used by CAMEO (<https://cameo3d.org/quality-estimation/>) for analyzing QA methods.

4.) Where are the predicted unreliable regions located? Is there any relationship between the secondary structures and the predicted unreliable regions?

We have added a new Supplementary Figure S5B, showing the breakdown of secondary structure types at the predicted unreliable regions. As shown in the figure, over 50% of unreliable regions are in loops, but the predicted regions are generally less accurate than preserved regions regardless of their secondary structure type.

5.) Can the authors report the accuracy of the final models (also see comment 2 above) selected with the highest predicted global IDDT (i.e., without any post-processing and/or structure averaging)?

We have added Figure S7 to show how DeepAccNet-Standard, DeepAccNet-Bert and DeepAccNet-MSA perform on model selection without structural averaging.

Minor Comments:

6.) Following the local bias removal process during unreliable region prediction by excluding contacts in the contact mask that are within sequence separation of 11, it

might be interesting to exclude contacts that are within sequence separation of 23 and focus only on long-range contacts.

We appreciate the reviewer's comment. We think there could be a better parameter set (like what the reviewer suggested) for unreliable region prediction that can be explored in future work.

7.) Line 274: typo? “..in then..”

Thank you for the suggestion that this is fixed now.

Reviewer #3 (Remarks to the Author):

This paper describes a deep learning approach to the problem of estimating the quality of protein models. The authors also explore how these estimates can be used to enhance the sampling of conformations during refinement using Rosetta. The paper is not that clearly written in several important areas, and I found it very hard to work out exactly how exactly the input features were derived, even after looking at the code.

We added comments to the featurization code as much as we can, especially focusing on the part where 3-D convolution inputs are generated. We hope this change helps users and developers understand better about our input featurization process.

The proposed approach is that of using 3-D convolutional units in a deep neural net to analyse the locations of atoms in the model being evaluated. This is not particularly novel, as several groups have already developed methods based on this same idea, and these papers have been cited in the text. The results presented in the paper are, however, good, certainly amongst the top-tier of methods in this general area, but these improvements are probably more down to using larger and/or better quality data sets, along with other improvements in the mechanics of how the networks were trained, than any methodological advances.

We believe that the use of additional deep 2D residual networks to tie 3-D convolutional units is novel among the QA methods. We would also like to point out that we make direct estimation of pairwise accuracy predictions rather than the just the per-residue accuracy estimation to make more direct coupling with refinement possible. We think that the newly added Figure4D proves this point. Additionally, during this revision round, we trained a version of DeepAccNet that uses a novel input feature that was not used in any other QA methods, which is sequence embeddings from the ProtBert-BFD100 model (referred to as

DeepAccNet-Bert in the manuscript), and showed that the feature considerably improves the performance of the network (Figure 3).

To our knowledge, our network is the first QA method to tie the aforementioned concepts, and our refinement framework is the first protocol that successfully applies deep learning to the protein refinement task and shows meaningful improvements.

As already mentioned, the overall results look good. I was a little unclear on whether there was any possibility of testing set contamination, however. CASP13 results feature prominently in the results shown, but I was surprised that the authors picked a cutoff date after the end of the CASP13 experiment for compiling their training data. Why not set the cutoff date just before the start of the original experiment? Surely that would be a more realistic comparator with other methods used in that experiment? I was also not sure whether the authors specifically excluded the CASP13 target structures that were released before Nov 2018 from their training data - certainly I can't immediately see any mention of them checking for this using something like ECOD, or did I just miss it?]

Thank you for pointing this out. We changed the cutoff date to the start of the CASP13 (05/01/2018). We adjusted the training, validation, and test set accordingly. We have re-trained all networks including the ones used to conduct feature analysis. We also re-ran the refinement analysis. All results and figures are updated accordingly.

Overall, we did not observe any significant change in our results.

The authors go to great lengths to separate methods which use multiple sequence alignments from those that don't, but I'm really not sold on their excuse for this. Some proteins won't have deep multiple sequence alignments, that's true. However, very few sequences are orphans given the current databank sizes, particularly with the availability of metagenomic sequences, which the Baker lab has already used in the past to good effect. They argue that protein design targets would not have alignments available, which is also true, but it's not clear to me how useful this study would be for protein design.

The main subject of the study is clearly protein structure prediction and modelling, not protein design. Certainly for protein modelling and refinement, there is no reason whatsoever to exclude multiple sequence alignments from use. The problem is that when MSA based methods are included in the benchmark, the results in supplementary material (Fig. S3) show that DeepAccNet is not appreciably better than some older methods which have traditionally done well at CASP e.g. ModFOLD. It might be unfair to say this, but I have to wonder if the true reason behind excluding those methods from the main text is to try to upsell the DeepAccNet results somewhat. Certainly, I believe that Figure S3 should be

included in the main text (Figure 2 can easily be dumped to supplementary material to make space if needed).

We agree with your point. We additionally trained two new models that include (i) MSA information in the form of inter-residue distance prediction from the trRosetta network and (ii) the protein sequence embedding from the ProtBert-BFD100 model. Figure 3 (CASP) and S4 (CAMEO) now compare the performance of DeepAccNet-Standard, DeepAccNet-Bert, and DeepAccNet-MSA to that of other methods, regardless of whether they rely on evolutionary information. Overall, DeepAccNet-Standard and DeepAccNet-Bert show superior performance compared to other methods that do not use evolutionary information. DeepAccNet-MSA shows state of the art performance compared to other methods that do use evolutionary information.

We believe that DeepAccNet can be useful for protein design. For example, one of the future research directions could be to use its predictions for assessing the quality of designed protein models. One way that protein design can fail is when modeled design structures do not actually closely resemble what they look like in nature. DeepAccNet can be potentially useful for filtering out bad designs.

The results on refinement seem reasonable enough. However, it's a shame that they did not test against another model quality method rather than just testing with and without their own method. I can see this would have been difficult to implement, though.

Following the reviewer's suggestion, we repeated two more refinement runs i) by replacing EMA method with VoromQA, which we think is one of the best single-model EMA methods available (also that most of the other EMA methods were very hard to install locally) or ii) by dropping the 2D estogram predictions from DeepAccNet. The parameters in the protocol using VoromQA were re-adjusted so that the fraction of predicted unreliable regions is similar to that by DeepAccNet. As shown in Figure 4D and Figure S5, the most critical contribution to the model refinement comes from the -D predictions derived from DeepAccNet estogram, while almost no differences were found in the other runs in which such precise 2D predictions were not available through EMA. This highlights the importance of 2D predictions by DeepAccNet for the refinement calculations which are not available yet from most other EMA tools.

I'm a bit concerned that the authors may not have excluded the effects of homology from their analysis here. Looking at Figure 4, a refinement jump from a GDT-HA score of 0.44 to 0.72 is massive – but I have to wonder if that target T0552 does actually have homologs in the training data. It's very hard for me to work this out given the numerous different splits and filter criteria the authors mention as they go along. Certainly, I would ask for firm evidence whether any of the cherry-picked examples in Figure 4 overlap _in any way_ with the data used to train DeepAccNet or which has been used for validation/parameter tuning.

We thoroughly investigated whether any homologous information would have affected the refinement performance during the revision. While curating the training set, any close homologue to the refinement test targets was excluded with a sequence identity over 35%. Then we also checked distant homologues that are structurally similar to the native structures of refinement examples. In the revised manuscript, Figure S4D is newly added to check the correlation between refinement performance and maximum sequence/structural similarity to the training set. We found almost no correlation in this analysis, showing the refinement occurs almost independent of the homologous information from the training set.

Finally, my main gripe. Feature selection is a key component of machine learning studies – homing in on only useful features makes ML models more robust and more importantly produces a more scientifically useful result. If there is smoke coming out of the engine of my car, I don't need to check the tire pressure or the color of the paintwork to work out where the problem is that needs fixing. In a good ML study, feature importance analysis is carried out quite early on, and only the useful features retained in the final study. Here, however, feature importance analysis has been left until the writing of the paper, and that is very poor engineering. The problem for me shows clearly in Figure 3A. What that appears to be telling me is that apart from the distances derived from the starting model, the only other features that are needed are the 3-D convolutions. The difference between plot i) and iii) looks close to zero to me. So basically, all the other stuff that the authors calculate and feed into the network simply aren't worth the effort of calculating them. So why isn't DeepAccNet just defined as being the model that takes the CB distances + 3D convolution? Adding the whole PyRosetta package as a code dependency, for example, is a huge complication (admittedly probably not for the Baker lab) when it's clearly not even needed. What's the point in all the extra wasted effort? These extra features make the work unnecessarily complicated, much harder to use and much harder to others to replicate and build upon. Unless the authors can show statistically that adding those extra features improves on results over the baseline, I strongly believe they should redefine their default approach to be that shown in Fig. 3Aiii – of course in that case the issue might remain as to how much novelty is left in the method, and whether it might be better dealt with in a more targeted journal e.g. Bioinformatics. However, I leave that for the Editors to decide.

Your point about the unnecessary complication with PyRosetta is certainly true. In this regard, we made the model (v; distance with 3D convolution) and the model (vii; distance with 3D convolution and Bert) available as well through the same github repository. These two networks do not require PyRosetta and we hope that it makes it easier for users to access our method.

We see a statistically significant difference between (ii) and (vii), though, suggesting that the features other than 3D-convolution and Bert help them glue together (Figure 3A, p-value < 0.0001 with Wilcoxon signed-rank test for estogram loss between network (ii) and (vii); MSA not included). We therefore also keep (i, ii, iii) so that users can decide what to use regarding the performance reported in the manuscript.

A final small point I would raise about the code & submission checklist. The authors state that the existing MIT license, which is fine, is "subject to change". I'm sorry, but I don't think that is acceptable. The authors need to decide under what license they are releasing the code needed to replicate the results of their study and stick to it. They are free to fork the code in the future, as is anybody else, but they must leave the current version of the code available as-is and under the same open source license it had when reviewed.

We will be releasing and keeping it under the MIT license.

REVIEWER COMMENTS

Reviewer #2 (Remarks to the Author):

Most of my concerns and comments were addressed. However, as a result of some the changes there are new significant changes that needs to be addressed, and in particular two key questions: whether the Bert information has any added value and how does DeepAccNet compare to the state-of-the-art model accuracy estimation methods? (more below)

DeepAccNet-Bert seems to outperform DeepAccNet-Standard on the test dataset (Figure 3), but DeepAccNet-Bert attains lower Spearman-r than DeepAccNet-Standard in the CAMEO dataset (Figure S4). Could the authors explain this discrepancy and add a significance test between the variants? Despite the considerable discussions on Bert embeddings, the main conclusion is basically that multiple sequence alignment (MSA) improves the performance for all test sets. Yet, MSA is an optional input to their network. While the authors argue that all available homology and co-evolutionary information is typically already used in generating the input models for protein structure refinement, the results are clearly showing significant contribution of MSA. Can the authors at least come up with some counterexamples to support their choice (i.e., protein targets where DeepAccNet-Standard performs better than DeepAccNet-MSA in model accuracy estimation and refinement) and present some case studies? In its current form I see no reason why one should not use MSA. No one would be happier than me if you could prove me wrong here.

Another major concern is the lack of comparison to the state-of-the-art single model accuracy estimation (EMA) methods. DeepAccNet and its variants should be compared to the recent EMA methods that utilize distance information such as ResNetQA (<https://doi.org/10.1101/2020.09.30.321661>) and QDeep (<https://doi.org/10.1093/bioinformatics/btaa455>) as well as to one of the top-performing EMA methods from CASP13: ProQ4. Without such comparisons, it is impossible to put DeepAccNet into context.

Reviewer #3 (Remarks to the Author):

I thank the authors for the work they have done addressing the points made by myself and the other reviewers. I still have a few niggles, but nothing worth going back to again. Overall the study and the paper are greatly improved and I would be perfectly happy to see it published as it is.

REVIEWER COMMENTS

Reviewer #2 (Remarks to the Author):

Most of my concerns and comments were addressed. However, as a result of some the changes there are new significant changes that needs to be addressed, and in particular two key questions: whether the Bert information has any added value and how does DeepAccNet compare to the state-of-the-art model accuracy estimation methods? (more below)

DeepAccNet-Bert seems to outperform DeepAccNet-Standard on the test dataset (Figure 3), but DeepAccNet-Bert attains lower Spearman-r than DeepAccNet-Standard in the CAMEO dataset (Figure S4). Could the authors explain this discrepancy and add a significance test between the variants? Despite the considerable discussions on Bert embeddings, the main conclusion is basically that multiple sequence alignment (MSA) improves the performance for all test sets. Yet, MSA is an optional input to their network. While the authors argue that all available homology and co-evolutionary information is typically already used in generating the input models for protein structure refinement, the results are clearly showing significant contribution of MSA. Can the authors at least come up with some counterexamples to support their choice (i.e., protein targets where DeepAccNet-Standard performs better than DeepAccNet-MSA in model accuracy estimation and refinement) and present some case studies? In its current form I see no reason why one should not use MSA. No one would be happier than me if you could prove me wrong here.

We apologize for the lack of clarity in our manuscript: we have made both the MSA and non-MSA methods available to the public and the user can "optionally" choose whichever they want based on their application. We found the non-MSA method more effective in evaluating de novo protein design quality, as there is no multiple sequence alignment available in this case.

To prove this point, we have additionally analyzed how DeepAccNet-variants performed on CASP14 targets that did not have any homologous sequence information. CASP14 targets were used because those are most independent of any training process. As reported in the newly added Figure S4, both Bert and Standard versions clearly outperformed the MSA version. As reviewer 2 pointed out, we noted in the revised main text that the MSA version would still be a more robust choice when multiple sequence alignment information is available in the revised manuscript.

Lastly, we added statistical significance tests between the DeepAccNet variants on the CASP13, CAMEO, and held-out test sets. Every test except for ones on the CAMEO global spearman-r values showed statistical significance at a significance level of 0.001 (Tables S3). The Bert version was not practically better than the standard version on the CAMEO set (Figure S5), and we believe this is because there was less room for practical improvements as the standard variant already did well on the set. On the other hand, for the CASP13 in which model quality assessment was generally harder, held-out test, and the refinement set, the addition of the Bert feature had a statistically and practically significant effect (Figure 4D). Thus, we believe that the Bert version still deserves discussion in the main manuscript.

Another major concern is the lack of comparison to the state-of-the-art single model accuracy estimation (EMA) methods. DeepAccNet and its variants should be compared to the recent EMA methods that utilize distance information such as ResNetQA (<https://doi.org/10.1101/2020.09.30.321661>) and QDeep (<https://doi.org/10.1093/bioinformatics/btaa455>) as well as to one of the top-performing EMA methods from CASP13: ProQ4. Without such comparisons, it is impossible to put DeepAccNet into context.

Thank you for the suggestion. CASP14 just concluded, and the methods described in this manuscript were top performers in both EMA and refinement categories. More than any other possible result, this puts DeepAccNet into context. Below is a figure taken from the EMA assessment presentation at CASP14 (Please find the link at the end of this comment).

DeepAccNet-standard was registered as "BAKER-experimental (group 403)", and DeepAccNet-MSA was registered as "BAKER-ROSETTASERVER (group 209)". Single model methods are shown in green.

This independent assessment shows that both DeepAccNet variants ("red rectangles") are top performers among single-model methods in both global and local QA. The DeepAccNet variants outperformed all potential competitors suggested by the reviewer ("blue rectangles").

https://www.predictioncenter.org/casp14/doc/presentations/2020_12_03_EMA_Assessment_Serok.pdf

Reviewer #3 (Remarks to the Author):

I thank the authors for the work they have done addressing the points made by myself and the other reviewers. I still have a few niggles, but nothing worth going back to again. Overall the study and the paper are greatly improved and I would be perfectly happy to see it published as it is.

Thank you for your comments. We agree and are happy that the manuscript is in much better shape.